# Household Financial Vulnerability to Income and Medical Expenditure Shocks: Measurement and Determinants

**DOI:** 10.3390/ijerph19084480

**Published:** 2022-04-08

**Authors:** Lei He, Shuyi Zhou

**Affiliations:** Hunan Key Laboratory of Macroeconomic Big Data Mining and Its Application, School of Business, Hunan Normal University, Changsha 410081, China; 202020030224@hunnu.edu.cn

**Keywords:** household financial vulnerability, ex ante measure, medical expenditure shock, health

## Abstract

This paper proposes a novel ex ante indicator to measure the degree of household financial vulnerability by calculating the probability of falling into financial distress under uncertain income and medical expenditure. The advantage of this measure is that it can reflect the capacity of households to deal with income shock and medical expenditure shock and quantify the degree of financial vulnerability for households beforehand. We employ it to measure the financial vulnerability of Chinese urban and rural households separately by using data from the China Health and Retirement Longitudinal Study (CHARLS) 2011, 2013, and 2015. Further, we analyze the potential determinants and their contributions to financial vulnerability changes. We find that rural households experience higher financial vulnerability than urban households. Furthermore, the investigation into the effects of potential determinants suggests that demographic variables (including age, gender, education, marital status, household size, labor force, and area), health-related variables (including health status, disability, and health shock), and medical insurance variables (including urban employee medical insurance and commercial medical insurance) have significant effects on the financial vulnerability of both urban and rural households. Contribution analysis of the determinants of household financial vulnerability shows that variables including disability, health shock, household size, labor force, and education contribute most to the changes in financial vulnerability.

## 1. Introduction

Since the recent financial crisis, the adverse financial situation of households has become the focus of financial institutions and central banks [1]. The term “household financial vulnerability” is suggested to describe the adverse financial situation that is linked to financial distress for indebtedness. The debt-servicing capacity of households affects the stability of financial institutions considerably [2]. It is confirmed that household financial vulnerability arising from indebtedness is closely related to the stability of financial systems [3,4]. The assessment of household financial vulnerability facilitates the reduction of the credit risk for financial institutions. Apart from this, household financial vulnerability reflects the failure to manage the cash flow efficiently and ensure resilience against economic shocks [5]. For these reasons, the measurement and analysis of household financial vulnerability are important for managing credit risk and preventing households from financial distress. The household financial vulnerability for some areas or countries are investigated by recent works, such as the Euro area [6], Indonesia [1], Chile [7], Pakistani [8,9].

There is little literature concerning household financial vulnerability in China. A rigid increase in household debt in China has been seen in recent years, out of which household residential mortgages have maintained rapid growth, short-term consumption loans have surged in 2017, and Internet finance has experienced rocket growth as a supplement to household funds [10]. With the persistent growth in housing prices over the past few decades across China, many households have undertaken high debt repayments for residential mortgages. From 2008 to 2017, the outstanding amount of residential mortgage loans have grown from RMB 3.0 trillion to RMB 21.9 trillion, taking up 45–54 percent of the total household debt. Due to market development, consumption upgrades, and the popularity of credit cards, the amount of short-term consumption loans have continued to rise in recent years, taking up to 6.8 trillion and increasing 19% year on year. Moreover, since 2013, Internet finance has developed dramatically by providing small-amount, short-term, and covenant-lite loans. It induces people to borrow excessive money beyond their repayment capacity. When households take on too much debt, their capacity to cope with uncertain income and unexpected expenditure will be weakened, and they become more vulnerable to adverse economic shocks. Medical expenditure shock is a critical source of poverty and financial distress in China because of the limited security level of the social medical insurance and expensive commercial insurance, especially for rural households. Therefore, the household financial vulnerability to medical expenditure shock needs to be paid more attention in China.

Those whose disposable income cannot cover the basic costs and loan installment payments are suggested to be vulnerable [2,6]. The sources of household financial vulnerability could be excessive indebtedness from consumption, income shock from unemployment, and medical expense shock from sudden diseases. In the existing literature, financial margin is the most popular indicator to measure household financial vulnerability. This indicator is defined as disposable income net of basic costs and loan installment payments [6]. It depicts households’ vulnerability to shocks of income and interest rate. To allow for the liquidity of assets, the definitions of household financial vulnerability are developed by [6,11].

However, medical expenditure is a major unanticipated payment for households [12]. The above definitions do not include the case that households fall into financial distress because of large, unexpected expenses incurred by disease shocks. The households who suffer more medical expenditure shock will have a higher probability of falling into financial distress and become more vulnerable. For some countries where households will suffer more unexpected medical expenditure because of a defective medical security scheme, traditional measures that only consider the income shock may underestimate the financial vulnerability. In order to assess or monitor effectively the financial distress risk of households for financial institutions and governments, a new measure is needed to reflect the capacity of households to cope with unexpected medical expenditures. Therefore, our motivation is to propose a measure of household financial vulnerability that not only includes the income shock, but also allows for the medical expenditure shock. To measure precisely the household financial vulnerability to income and medical expenditure shocks by a probability method, the dependence of income and medical expenditure should be rigorously depicted.

Inspired by the definitions of [6,12], our paper defines financially vulnerable households as those who are easy to fall in financial distress due to uncertain income and unexpected medical expenditures, namely, households that have negative financial margins and do not have enough liquid assets to cover unexpected medical expenditures. Based on this definition, we propose a method to measure the degree of household financial vulnerability under the uncertainty of income and medical expenditures, using the probability that financial margin is negative and liquid assets are less than medical expenditures. The process is designed as follows. We first construct the margin distributions of income and medical expenditure conditional on household characteristics by estimating two equations with idiosyncratic shocks. Subsequently, the joint distribution of income and medical expenditure is built by introducing a copula function to capture the dependence between the two shocks. Given the joint distribution of income and medical expenditure, the probability of falling into financial distress conditional on household characteristics can be calculated.

The above method is applied to investigate the financial vulnerability of households in China by using a panel dataset obtained from the China Health and Retirement Longitudinal Study (CHARLS), which provides three waves (2011, 2013, and 2015). Firstly, we assess the financial vulnerability of households conditional on household characteristics. Considering the considerable economic disparity between Chinese urban and rural areas, the financial vulnerability is calculated separately for urban and rural households. Secondly, we analyze the potential determinants of household financial vulnerability and find that demographic variables (including age, gender, education, marital status, household size, labor force, and area), health-related variables (including health status, disability, and health shock), and medical insurance variables (including urban employee medical insurance and commercial medical insurance) have significant effects on financial vulnerability for both urban and rural households. Finally, we investigate the contribution of each determinant to the changes in household financial vulnerability during 2011–2013 and 2013–2015. It is found that variables including disability, health shock, size, labor force, and education of the head of the household contribute most to the changes of financial vulnerability in urban households. Moreover, household size, labor force, and education can account for most of the changes in financial vulnerability in rural households, while medical insurances contribute little.

Our study contributes to the literature as follows. First, we introduce a probability method to measure household financial vulnerability under uncertain income and medical expenditure. This measure provides an ex ante indicator of household financial vulnerability considering income and medical expenditure shocks. The ex ante indicator of household vulnerability can mostly be found in literature regarding poverty vulnerability, which is measured by the probability of consumption or income being less than the poverty line [13,14]. Differing from them, we propose the ex ante indicator of household financial vulnerability by using the probability that financial margin is negative and liquid assets are less than medical expenditures when income and medical expenditure are uncertain. It depicts the probability of a household falling into financial distress due to facing income and medical expenditure shocks. Moreover, although there is literature that argues income and health shocks as determinants of financial vulnerability, most of them focus on examining the casual relationship between household financial vulnerability and medical expenditure shocks or income shocks. The income and medical expenditure shocks are not included in the indicators of financial vulnerability. Second, our measure method allows for the dependence of income and medical expenditure shocks by introducing a copula function to depict the dependence of them. Differing from the ex ante measure of [15], the heteroscedasticity of income and medical expenditure shocks for different characteristics of households is included in our measure, and it is estimated by a three-step feasible generalized least squares (FGLS) procedure. It is conducive to precisely assessing the financial vulnerability of a household based on the characteristics of that household. Third, we employ a novel measure to assess household financial vulnerability in the context of China using the Chinese CHARLS survey data and investigate the determinants of China’s household financial vulnerability.

The paper is organized as follow. The next section offers a brief overview of the literature. Section 3 proposes the theoretical framework regarding the definition and measure approaches of household financial vulnerability. Section 4 describes the dataset and variables and shows some descriptive statistics. The empirical results are presented in Section 5 and are discussed in Section 6. Section 7 concludes the article.

## 2. Literature Review

There are three streams of literature related to our work: (1) definitions of household financial vulnerability; (2) measurement of household financial vulnerability; (3) the determinants of it. There is no universally accepted definition of household financial vulnerability in the existing literature. This term is used interchangeably with financial distress, financial fragility, financial debt burden, and overindebtedness [16].

The definition of household financial vulnerability based on debt burden is the most frequently employed in the existing studies. Overborrowing may lead to debt level being too high relative to the capacity of repayment [17]. When focusing on the default risk of households, household financial vulnerability is defined as the failure of a household to repay its debt [18,19]. The definition proposed by [17] takes into account not only over-commitment because of excess indebtedness, but also the inability to meet monthly expenses for food items, to balance the budgets, and to pay the rent and utility bills. Ref. [7] defines as financially vulnerable households those whose probability of being excessively indebted exceeds a corresponding threshold. When focusing on the liquidity risk of portfolio allocation held by households, Ref. [11] defines financially vulnerable households as those whose income can cover expected expenses but do not have sufficient liquidity buffers to meet unexpected ones. In addition, Ref. [12] argues that households who are unable to cope with ordinary financial shocks are financially vulnerable. The ability to survive in the event of economic or financial shocks is also included in the definition of financial vulnerability from [16]. In the existing literature, the indicators for measuring financial vulnerability are mainly divided into two types, which are objective indicators and subjective indicators. The objective indicators mainly include ratios regarding debt burden and financial margin. The ratios concerning debt burden include the debt-service to income ratio [6,8,9,18,20,21,22], debt-to-income ratio [6,8,23], and debt-to-asset ratio [6,8,24]. Although these ratios can reflect the level of indebtedness, they require an arbitrary threshold to identify the financially vulnerable households. For example, [22] selects a threshold of 30% in the debt-service to income ratio. Ref. [20] argues that households with a debt-service to income ratio greater than 40% are vulnerable. However, the selection of these thresholds lacks an exact standard.

To address the drawback of the above indicator of debt burden ratios, the financial margin is proposed to measure the households’ financial vulnerability. It is defined as disposable income net of basic living costs and installment loans [25]. Households with a negative financial margin are suggested to be vulnerable [6,26]. Taking into account the liquidity position of households, Ref. [6] measures financial vulnerability by a condition in which the negative financial margin is greater than the liquid assets during a certain period. This indicator is also used to empirically investigate the financial vulnerability of Italian households in [27]. The advantage of financial margin is that it is suitable for assessing households, regardless of whether they have debts or not [8]. However, this indicator is flawed for some countries where the level of indebtedness is low, and a very limited proportion of households with negative wealth are found, such as Italy [11].

In addition to the above-mentioned objective indicators, subjective indicators are also suggested to measure household financial vulnerability in some of the literature. Ref. [12] measures household financial fragility by a self-assessed indicator that reflects households’ capacity to deal with financial shocks. Ref. [17] employs a subjective indicator based on some self-reported variables by a series of questions to measure financial vulnerability. Similarly, Ref. [16] measures the households’ financial vulnerability by three questions that, respectively, reflect the inability to meet household needs, the inability to survive, and the inability to cope with unexpected expenses. Ref. [7] makes use of households’ self-assessed debt burden to measure the financial vulnerability of households arising from consumer debt. Moreover, Ref. [1] proposes a mixed index of financial vulnerability based on both objective indicators and subjective indicators, in which the objective indicators evaluate the financial condition of households in monetary terms, and the subjective indicators assess the financial condition based on household expectations, feelings, or perceptions. Most objective and subjective measures are ex post measures that cannot measure the probability of becoming vulnerable in the future for those who are not vulnerable at present. The ex ante measure of household vulnerability is mainly applied to investigate the vulnerability of households to poverty by the probability method. Few ex ante measures of household vulnerability have been applied in empirical analysis. For example, Ref. [15], which proposes an ex ante measure of household financial fragility by the probability of income being less than expenditure.

Based on the measurements, the investigations into the determinants of household financial vulnerability are performed by using different survey data. The empirical analysis of [17] points out that household debt servicing, impulsive emotion, and financial literacy are significant determinants of household vulnerability when controlling the demographic characteristics based on Italian data. Ref. [11] provides evidence of homeownership being an important factor of financial fragility other than the usual socio-demographic features for Italian households. The results of [16] show that the significant determinants of Malaysian households’ financial vulnerability include financial behavior in money management, level of education, income level, marital status, and age. Ref. [1] finds that the financial vulnerability of Indonesian households is significantly affected by finance-related behavioral characteristics (including the number of assets and the ownership of financial instruments) and social-economic factors (including supply of household labor and educational level) other than income factors. Ref. [7] shows that age, being married, the number of people in the household, and years of education have statistically significant effects on the financial vulnerability of households in Chile. Ref. [9] documents empirically that financial vulnerability declines with increasing household heads’ education by using survey data in Pakistan, and they find other significant determinants of financial vulnerability, including gender, region, marital status, province, and employment status of household heads.

## 3. Methodology

As the most popular indicator to measure household financial vulnerability in the existing literature, financial margin, which is defined as disposable income net of basic costs and loan installment payments, depicts household vulnerability to shocks of income and interest rate. However, the asset allocation of existing wealth is not considered in this measurement [6]. The liquidity of household wealth is another important issue in household finance. It determines the capacity of households to cover the negative financial margin and unexpected expenditures. Therefore, those whose financial margin is negative and liquid assets are unable to cover the negative financial margin are considered to be financially vulnerable by [6]. Moreover, Ref. [11] also includes the inability to pay for unexpected expenditures in the measurement of financial vulnerability. They argue that a household becomes vulnerable when its income is more than expected expenses and its liquid assets are less than its unexpected expenses.

Inspired by the definition of [6,11], we define as financially vulnerable households who have negative financial margin and do not have enough liquid assets to cover their unexpected expenses. This definition emphasizes the situation of financial distress and illiquidity of wealth. Usually, expenditures of medical emergencies are the major source of unexpected expenses. In order to consider the ability of households to deal with this shock, we further specify that households with negative financial margin and liquid assets being less than medical expenditures are vulnerable. Differing from the ex post and static measures of household financial vulnerability using dummy variables, we propose an ex ante measure based on the probability that households face a negative financial margin and have fewer liquid assets than medical expenditures because of income and medical expenditure shocks. Formally, the degree of *i*-th household’s financial vulnerability, $\Delta_{i,}$, is measured by following probability:(1)$$\Delta_{i,}=P\left({\mathrm{FM}}_{i}\;{<0,\mathrm{LA}}_{i}\;{<\mathrm{ME}}_{i}\right)$$
where ${\mathrm{FM}}_{i}$, ${\mathrm{LA}}_{i}$, and ${\mathrm{ME}}_{i}$ are financial margin, liquid assets, and medical expenditures, respectively.

According to [6], the financial margin is calculated by:(2)$${\mathrm{FM}}_{i}{=I}_{i}-{\mathrm{DP}}_{i}-{\mathrm{BLC}}_{i}$$
where $I_{i}$, ${\mathrm{DP}}_{i}$, and ${\mathrm{BLC}}_{i}$ are disposable income, debt payments paid by household, and basic living costs, respectively. Because of income shocks and medical expenditure shocks, $I_{i}$ and ${\mathrm{ME}}_{i}$ are random variables and the asset allocation, debt payments, and basic living costs are assumed to be constant in the short term.

In most related literature, income and consumption are performed by log transformation in regression and are assumed to be log-normally distributed random variables, such as [28,29,30]. We assume that disposable income, $I_{i}$, and medical expenditure, ${\mathrm{ME}}_{i}$, are governed by the following regression equations:(3)$$lnI_{i}={\alpha X}_{i}+\varepsilon_{I,i},\varepsilon_{I,i}\sim N(0,\sigma_{I,i}^{2})$$
(4)$$ln{\mathrm{ME}}_{i}=\beta Y_{i}+\varepsilon_{\mathrm{ME},i},\varepsilon_{\mathrm{ME},i}\sim N(0,\sigma_{\mathrm{ME},i}^{2})$$
where $X_{i}$ represents a set of observable household characteristics that determine the disposable income and $Y_{i}$ represents a set of the characteristics that affect the medical expenditure. α and β are vectors of parameter, and $\varepsilon_{I,i}$ and $\varepsilon_{\mathrm{ME},i}$ are the idiosyncratic shocks regarding income and medical expenditure, respectively. $E_{I,i}$ and $\varepsilon_{\mathrm{ME},i}$ are also interpreted disturbance terms and are assumed to distribute normally with zero mean and variances $\sigma_{I,i}^{2}$ and $\sigma_{\mathrm{ME},i}^{2}$, respectively, i.e., $\varepsilon_{I,i}\sim N\left(0,\sigma_{I,i}^{2}\right)$, $\varepsilon_{\mathrm{ME},i}\sim N\left(0,\sigma_{\mathrm{ME},i}^{2}\right)$.

To address the problem of the above heteroscedasticity, traditional parametric models are suggested where the variances $\sigma_{I,i}^{2}$ and $\sigma_{\mathrm{ME},i}^{2}$ are related to the sets of observable household characteristics $X_{i}$, and $Y_{i}$. To simplify, we select the following models:(5)$$\sigma_{I,i}^{2}=\theta X_{i}$$
(6)$$\sigma_{\mathrm{ME},i}^{2}=\omega Y_{i}$$

Given the household characteristics $X_{i}$ and $Y_{i}$, the conditional marginal distributions of $lnI_{i}$ and $ln{\mathrm{ME}}_{i}$ can be denoted by $G\left({u;\alpha ,\theta |X}_{i}\right)$ and $H\left({v;\beta ,\omega |Y}_{i}\right)$, respectively, and the corresponding density functions are $g\left({u;\alpha ,\theta |X}_{i}\right)$ and $h\left({v;\beta ,\omega |Y}_{i}\right)$.

Further, considering the exiting dependence between income shock and medical expenditure shock, we assume that $lnI_{i}$ and $ln{\mathrm{ME}}_{i}$ have a joint distribution function, F(u,v). In order to depict the dependence between the two variables, the copula function is introduced. According to Sklar’s theorem, the joint distribution function can be specified by a unique copula function and marginal distributions, i.e.,
(7)$$F\left(u,v\right)=C\left(G\left({u;\alpha ,\theta |X}_{i}\right),H\left({v;\beta ,\omega |Y}_{i}\right);\gamma \right)$$
where γ is the parameter of copula function. The copula function can be selected by a goodness-of-fit test, and its parameter, γ, can be estimated by the maximum likelihood estimation. Then, the joint density function, $\left(lnI_{i},ln{\mathrm{ME}}_{i}\right)$, is:(8)$$f\left(u,v\right)=c\left(G\left({u;\alpha ,\theta |X}_{i}\right),H\left({v;\beta ,\omega |Y}_{i}\right);\gamma \right)g\left({u;\alpha ,\theta |X}_{i}\right)h\left({v;\beta ,\omega |Y}_{i}\right)$$

Next, we can calculate the degree of household financial vulnerability for the household with characteristics $X_{i}$ and $Y_{i}$ by following double integral:(9)$$\begin{array}{cc} \Delta_{i}\left(X_{i},Y_{i}\right) & =P(lnI_{i}<ln\left({\mathrm{DP}}_{i}+{\mathrm{BLC}}_{i}\right),ln{\mathrm{LA}}_{i}<ln{\mathrm{ME}}_{i}{|X}_{i},Y_{i})\\ & =\int_{-\infty }^{ln\left({\mathrm{DP}}_{i}+{\mathrm{BLC}}_{i}\right)}\int_{ln{\mathrm{LA}}_{i}}^{+\infty }c\left(G\left({u;\alpha ,\theta |X}_{i}\right),H\left({v;\beta ,\omega |Y}_{i}\right);\gamma \right)\\ & \;\;\;\;g\left({u;\alpha ,\theta |X}_{i}\right)h\left({v;\beta ,\omega |Y}_{i}\right)\mathrm{dvdu} \end{array}$$

The household vulnerability measure can be calculated by the following steps.

Firstly, we specify the marginal distributions of log income and log medical expenditure and estimate α, β, θ, and ω in Equations (3) and (4) using a three-step feasible generalized least squares (FGLS) procedure suggested by [31]. By this method, we are able to directly estimate the expected log income and the expected log medical expenditure:(10)$$\hat{E}[ln\;I_{i}|X_{i}]=\hat{\alpha }X_{i}$$
(11)$$\hat{E}[ln\;ME_{i}|Y_{i}]=\hat{\beta }Y_{i}$$
and the variance of log income and the variance of log medical expenditure:(12)$$\hat{V}[ln\;I_{i}|X_{i}]={\hat{\sigma }}_{I,i}^{2}=\hat{\theta }X_{i}$$
(13)$$\hat{V}[ln\;ME_{i}|Y_{i}]={\hat{\sigma }}_{\mathrm{ME},i}^{2}=\hat{\omega }Y_{i}$$

Then, we get $lnI_{i}|X_{i}\sim N(\hat{\alpha }X_{i},\hat{\theta }X_{i})$ and $ln\;ME_{i}|Y_{i}\sim N(\hat{\beta }Y_{i},\hat{\omega }Y_{i})$.

Secondly, we select the copula function, C(u,v;γ), by goodness-of-fit test, and estimate the parameter $\hat{\gamma }$ by maximum likelihood estimation.
(14)$$\hat{\gamma }=argmax\sum ln\;c\left(G\left({u;\hat{\alpha },\hat{\theta }|X}_{i}\right),H\left({v;\hat{\beta },\hat{\omega }|Y}_{i}\right);\hat{\gamma }\right)$$

Thirdly, the financial vulnerability of household $\Delta \left(X_{i}{,Y}_{i}\right)$ can be calculated using Equation (9).

## 4. Data and Descriptive Statistics

Based on the above measurement, we estimate the household financial vulnerability to income and medical expenditure shocks in China using the dataset drawn from the China Health and Retirement Longitudinal Study (CHARLS), which provides three waves (2011, 2013, and 2015). The raw data used in the present study is publicly available at http://charls.pku.edu.cn/en/ (accessed on 6 June 2020). The dataset includes a total of 14,277 households, including 5559 urban households and 8718 rural households.

Under the CHARLS surveys, we collect economic-financial indicators at the household level, including income per capita, medical expenditures per capita, debt payments per capita, and liquid assets per capita. We also collect information on socio-demographic indicators, including age; gender; education years; marital status of household head; household size; labor force and area; and health-related variables such as health status, chronic diseases, disability, and health shock. The medical insurance related variables are also considered, such as urban employee medical insurance, commercial medical insurance, and other medical insurance. Other medical insurance mainly refers to the residents’ medical insurance, which is provided to people without employee medical insurance. The definitions of indicators are reported in Table 1. Following [9], the basic living cost is calibrated by referring to the international standard poverty line, which is USD 2 per day per person.

The descriptive statistics of the variables are reported in Table 2. Considering the difference between urban and rural areas in China, all the variables are summarized separately for urban and rural households. The mean disposable income per capita of urban households is CNY 8839.73 a year, while that of rural households is CNY 6963.38 a year. It reflects that there is still a certain income gap between urban and rural residents in China. The average medical expenditures per capita of urban and rural households are CNY 1477.64 and CNY 1518.05 a year, respectively. The mean debt payments per capita for urban and rural households are CNY 107.65 and CNY 66.10, respectively. The liquid assets per capita owned by households in urban and rural areas are CNY 5339.31 and CNY 3591.45, respectively. The average age of household heads in urban and rural areas is approximately 61 years old and the average education years of household heads are approximately 6 for both urban and rural areas. Nearly 80% of the household heads are male and nearly 80% of the heads are married in both urban and rural areas. There is not much difference in household size between urban and rural households, which are 3.90 and 3.88, respectively. The labor force measures the proportion of household members at working age. The means of it are 39% and 37% for urban and rural households, respectively.

With regard to health status, the mean value of the self-reported health status of household heads is approximately 4, i.e., fair. Chronic disease as a binary variable is used to measure whether the household head or spouse has been chronically ill in the past two years. The means of it show that most of them (i.e., 86% of urban households and 89% of rural households) have experienced chronic illness. Health shock is used to measure whether the household head or spouse has been hospitalized. The means of it show that 30% of household heads or spouses have been hospitalized. In terms of medical insurance, China’s basic medical insurance coverage rate is relatively high. In urban samples, households with employee medical insurance account for 19%, and households with the other medical insurance account for 79%, while they account for 11% and 86% in the rural samples, respectively. However, the urban households participating in commercial medical insurance account for only 4%, and the rural households account for only 3%. The average values of these economic variables show that there is a significant difference between urban and rural areas. Therefore, we measure the financial vulnerability of urban and rural households separately.

## 5. Empirical Results

### 5.1. Household Financial Vulnerability

To calculate the household financial vulnerability to income and medical expenditure shocks, we first estimate the income and medical expenditure equations. The independent variables of the income equation include age, gender, education, married, size, labor, area, disabled, and health_S. The independent variables in the medical expenditure model are not exactly the same as the income regression model. The medical insurance variables, including insur_EM, insur_CM, and insur_OM, are also taken into account. To specify the marginal distributions of income and medical expenditure, we estimate parameters in income and medical expenditure equations using a three-step feasible generalized least squares (FGLS). The estimated parameters are reported in the Appendix A. We then choose an appropriate copula function to capture the dependence of income and medical expenditure shocks. The goodness-of-fit test results show that the normal copula is an appropriate choice to capture the dependence between income shock and medical expenditure shock. At the same time, the association parameter of the urban samples is 0.144 and rural is 0.214, which indicate a positive dependence between income and medical expenditure variables. Therefore, we can specify the joint distribution of income and medical expenditure. Subsequently, we calculate the probability of households falling into financial distress according to Equation (9). The household financial vulnerability of different groups is summarized in Table 3, Table 4 and Table 5.

Table 3 reports the average financial vulnerability of groups with different health status of the household heads in 2011, 2013, and 2015. The average financial vulnerabilities of urban household heads with excellent, very good, good, fair, poor, or very poor health status in 2011 are 0.000, 0.025, 0.038, 0.051, 0.071, and 0.099, respectively. This shows that households become more financially vulnerable when their health status is poorer. We also find that the financial vulnerability of households increased during the period 2015 to 2011. These phenomena can also be found in rural household groups. However, rural households are more financially vulnerable than urban households. This further indicates that rural households are more likely to fall into financial distress than urban households.

The average values of the financial vulnerability of groups with different labor force proportions are presented in Table 4. It is found that households with a low labor force proportion are more likely to become financially vulnerable, whether in urban areas or in rural areas. For example, in 2015, the financial vulnerabilities of rural household groups with labor force proportion being less than 0.25, 0.25–0.5, 0.5–0.75, and more than 75, are 0.132, 0.113, 0.062, and 0.070, respectively. The household financial vulnerability of each group has also increased during the period 2011 to 2015. Moreover, the household financial vulnerability of groups in rural areas is higher than that in urban areas.

We report the average financial vulnerability of households with different education years of heads in Table 5. Households headed by people with less than six years of education are most likely to fall into financial distress, and the longer the education years of the household head, the lower the financial vulnerability is faced by urban households. For example, in 2015, the financial vulnerabilities of urban households with heads experiencing less than 6, 6–9, 9–12, and 12–15, and more than 15 years of education are 0.155, 0.119, 0.096, 0.079, and 0.047, respectively. A similar phenomenon can be seen in rural households.

### 5.2. Determinants of Household Financial Vulnerability

On the basis of household financial vulnerability measured by the probability of falling into financial distress, we analyze the determinants of it for urban and rural households. We consider the demographic variables, such as age, gender, education, marital status, household size, labor force, and area. The health-related variables, such as health status, chronic diseases, disability, and health shock, are also included. Medical insurance is another important factor of household financial vulnerability. This is represented by urban employee medical insurance, commercial medical insurance, and other medical insurance. The set of independent variables is typically used in the analysis of household financial distress. The Chow test is conducted to examine the difference of models for 2011–2013 and 2013–2015 data. The null hypothesis cannot be rejected. Following [17,32], we analyze the determinants of financial vulnerability through pooled OLS (Ordinary Least Squares), fixed effect, and random effect estimators. The results are presented in Table 6.

We perform the F test for individual effect and conduct the Hausman test based on the results of fixed effect and random effect estimators. The results show that there is an individual effect in the panel data model, and the hypothesis of no fixed effect cannot be rejected. We focus on the results of the fixed effect model. Amongst the demographic determinants, the coefficients of education, household size, labor force, and area are significant for urban households. The coefficients of demographic determinants are significant for rural households. Although the age of the head negatively affects the financial vulnerability of rural households, it has no significant effect on the financial vulnerability of urban households. The results regarding the age variable for rural households are in accordance with [1], where the financial vulnerability of Indonesia is assessed by a subjective measures. It indicates that rural households with younger heads are more vulnerable than those with older heads. This may be because households with older heads are more conservative in their financial behavior and thus more stable [11,16]. The coefficient of gender is significantly positive for rural households, but not significant for urban households. This finding contradicts [8], which uses several ex post measures to assess the financial vulnerability. As an important determinant, the level of education helps to reduce the financial vulnerability of households, which is consistent with the finding of Pakistan household financial vulnerability in [9]. Coinciding with Italian households in [17], marital status plays a significant role in impacting vulnerability levels. The marital status of rural households has a significantly negative impact on financial vulnerability, while the significance of it is not found for urban households. Similar results are found in Chile by [7], where a financially vulnerable household is defined by the probability of being highly or excessively indebted above a threshold. Large household size and low proportion of the labor force significantly increase the vulnerability of households, as do the findings of [1]. The area variable refers to whether the household is located in eastern China, which is a more developed region. The results show that households living in the eastern region face lower financial vulnerability. This can be explained by the fact that there is less income shock because of sufficient employment opportunities in the eastern region.

As for health-related variables, the health status of household head, disability of head or spouse, and health shock have significant effects on the vulnerability levels in both urban and rural households. Poor health status of the household head, generally associated with unexpected income loss and/or higher medical expenses, increases the level of financial vulnerability. Disability of the head or spouse limits the disposable income of the household and leads to high vulnerability. Chronic disease of the head or spouse has no significant effect on urban households, whereas it has a significant effect on rural households. The reason for this may be that urban households enjoy more comprehensive social security than rural households. Medical insurance is a key tool to manage the medical expenditure risk for households. The results show that employee medical insurance or commercial medical insurance can significantly reduce household financial vulnerability to some extent, while commercial medical insurance and other medical insurance has no significant effect on household vulnerability. Moreover, commercial medical insurance significantly affects the financial vulnerability of rural households. Other medical insurance, mainly referring to resident medical insurance, provides low reimbursement for medical expenditure but asks for increasing premium. Therefore, its effect on household financial vulnerability is ambiguous.

### 5.3. The Contribution of Determinants to the Change of Household Financial Vulnerability

Following the method of [33], we can further analyze the contribution of various components to the change of household financial vulnerability by measuring the elasticity and the rate of change. Table 7 reports the absolute and relative contributions of each determinant to the changes in financial vulnerability for urban households in the periods 2011–2013 and 2013–2015. From this, we can find that the financial vulnerability of urban households declined by 19.30% during the period 2011–2013. According to the results of relative contribution, 17.97% of the decline comes from the decrease in the household size, and 15.74% comes from the increase in the proportion of household labor. In addition, due to the promotion of government policy, households with urban employee medical insurance have a great increase from 2011 to 2013, which contributes 57.14% to the decline of household financial vulnerability. The decline in the education years of household heads has a negative effect of 12.77% on the decline of household financial vulnerability. During the period 2013–2015, the financial vulnerability of urban households increased by 71.74%. The demographics variables (including size, labor force, and education years of household head) and the health characteristics variables (including disable and health shock) have high relative contributions among the determinants.

Table 8 reports the absolute and relative contributions of each determinant to the changes in financial vulnerability for rural households in the periods 2011–2013 and 2013–2015. From 2011 to 2013, the financial vulnerability of rural households declined by 17.28%. As can be seen from Table 8, the decrease in household size contributes 22.73% to the decline, and the increase in the proportion of household labor contributes 13.43%. In contrast to the urban samples, the education years of the rural household heads increased by 23.10% in 2013 and contributed 46.44% to the decline in household financial vulnerability. During the period 2013–2015, the financial vulnerability of rural households increases by 52.24%, where household size and labor force have high relative contributions among the determinants.

## 6. Discussion

This study focuses on introducing a novel method to measure household financial vulnerability under uncertain income and unexpected medical expenditure. Our study contributes to the literature by proposing a probability method to assess household financial vulnerability to uncertain income and unexpected medical expenditures and applies it to investigate household financial vulnerability in the context of China. Differing from the ex post indicators that measure household financial vulnerability in existing studies (e.g., [6]), we propose an ex ante indicator to measure the degree of household financial vulnerability by calculating the probability of falling into financial distress with negative financial margin and deficient liquidity assets. To capture the vulnerability of households to the uncertainty of income and medical expenditure, the marginal distributions of income and medical expenditure are assumed to be lognormal conditional on household characteristics. The parameters of conditional marginal distributions are estimated by the three-step feasible generalized least squares method. Meanwhile, a copula function is estimated to depict the dependence between the income and medical expenditure shocks. The goodness-of-fit test results show that the normal copula is an appropriate choice. On the basis of the estimated conditional marginal distributions and copula function, we can obtain the joint distribution of income and medical expenditure conditional on the household characteristics. Then, the financial vulnerability for each household is measured by calculating the probability of falling into financial distress. The highlights of this measure method are manifested in two aspects. On the one hand, it can reflect the capacity of household to deal with income and medical expenditure shocks. On the other hand, the probability of falling into financial distress can quantify the degree of household financial vulnerability beforehand, which helps to assess the financial risks of the household sector. The above measure is applied to investigate household financial vulnerability in the context of China. The financial vulnerability of urban and rural households in China is measured separately for the considerable economic disparity between urban and rural areas. The descriptive statistical analysis shows that rural households experience higher financial vulnerability than urban households. The reason for this may be that the rural households face more income shocks and medical expenditure shocks and have a lack of risk management instruments. Then, we estimate the effects of potential determinants on household financial vulnerability. It is found that demographic variables (including age, gender, education, marital status, household size, labor force, and area), the health-related variables (including health status, disability, and health shock), and medical insurance variables (including urban employee medical insurance and commercial medical insurance) have significant effects on financial vulnerability for both urban and rural households.

However, there are some differences between urban and rural households. First, the age of the head negatively affects the financial vulnerability of rural households, whereas it has no significant effect on the financial vulnerability of urban households. The reason for the difference in significance between urban and rural households is that younger labor can obtain more income in the agricultural industry for rural households, while the income of urban households mainly depends on the occupation and is weakly affected by age in China. Second, the gender of the household’s head has a significantly positive effect on the financial vulnerability of rural households, but there is no significant effect of urban households. This can be explained by the fact that male labor can obtain more income than female labor in agricultural production, and households with a male head may suffer little income shock. Third, the marital status of rural households has a significantly negative impact on financial vulnerability, while the significance of it is not found for urban households. Rural households with a married head experience lower income per capita and higher living costs than a single head in China. Thus, rural households with a married head may be more financially vulnerable because of low savings. Fourth, disease has a significant effect on the financial vulnerability of rural households, whereas it has an insignificant effect on urban households. This is caused by the fact that rural households own fewer health resources than urban households in China. Moreover, the effect of commercial medical insurance on the financial vulnerability of rural households is higher than that of urban households. As an important tool for health risk management, the other medical insurance for rural households provides lower reimbursement than the employee medical insurance owned by most urban households. Thus, rural households rely more on commercial medical insurance that can provide higher reimbursement.

Further research is performed to investigate the contribution of determinants to the changes in household financial vulnerability during the period 2011–2015 by using the method of [33]. For urban households, financial vulnerability first declined from 2011 to 2013, and then increased from 2013 to 2015. The health characteristics variables (including disability and health shock) and the demographic variables (including size, labor force, and head education) have been the most significant contributors to these changes. The demographic variables, which include size, labor force, and education of the head, can mostly account for similar changes in financial vulnerability in rural households. Moreover, medical insurances contribute little to changes in financial vulnerability in rural households. Expensive premiums for commercial medical insurance and low reimbursement of social medical insurance for rural households in China can account for this phenomenon.

## 7. Conclusions

In this paper, we propose a novel ex ante indicator to measure the degree of household financial vulnerability by calculating the probability of falling into financial distress under uncertain income and medical expenditure. An excellent feature of this is that it can reflect the capacity of households to deal with income shock and medical expenditure shock beforehand. Using the measurement, we first calculate the household financial vulnerability to income and medical expenditure shocks in China using a panel dataset drawn from the China Health and Retirement Longitudinal Study (CHARLS). Then, we estimate the effects of potential determinants on household financial vulnerability. Finally, the contributions of determinants to changes in financial vulnerability for urban and rural households are investigated.

Household financial vulnerability is measured by the probability of households falling into financial distress. We find that rural households are more financially vulnerable than urban households in China. Our investigation shows that demographic variables, health-related variables, and medical insurance variables have significant effects on the financial vulnerability of both urban and rural households. The effects of these determinants present the idiosyncratic characteristic because of the considerable disparity in economic development and social security between Chinese urban and rural areas. The financial vulnerability of Chinese households first declined from 2011 to 2013, then increased more from 2013 to 2015. Variables including disability, health shock, size, labor force, and education of head contribute most to the changes of financial vulnerability.

Our work has some policy implications and practical value. Promotion of education and employment creation can decrease the financial vulnerability of households by lowering the income shock in China, especially for rural households. Health shock contributes mainly to the financial vulnerability of Chinese households. However, as risk management tools, medical insurances have made few contributions to the changes in financial vulnerability for the defective medical insurance system in China. Due to a lack of formal employment opportunities, which provide the employee medical insurance with more coverage and reimbursement, rural households exhibit more financial vulnerability. Thia implies that the medical insurance policy for Chinese rural households needs to be improved by increasing coverage and reimbursement of social insurance, especially for serious illness. In the way of practical value, our measure facilitates the financial institutions to assess the default risk of households being in financial distress before loan approval, and it is also conducive to identifying households with high financial vulnerability, and monitoring their evolution.

## Figures and Tables

**Table 1 ijerph-19-04480-t001:** Variables and their description.

Variable	Symbol	Description and Measure
Income	I	Household disposable income per capita
Medical expenditure	ME	Household medical expenditure per capita
Debt payments	DP	Household debt payments per capita
Liquid assets	LA	Household liquid assets per capita
Age	Age	Age of household head in years
Gender	Gender	Gender of household head (male = 1, female = 0)
Education	Education	Years of education of household head
Marital status	Married	Marital status of household head (presently married = 1, single/divorced/separated/widowed = 0)
Household size	Size	Household size
Labor force	Labor	Proportion of members at working age
Area	Area	Whether located in the eastern region of China (yes = 1, no = 0)
Health status	Health	Health status of household head (self-report) (excellent = 1, very good = 2, good = 3, fair = 4, poor = 5, very poor = 6)
Chronic diseases	Diseases	Whether household head or spouse has been chronically ill in the past two years (yes = 1, no = 0)
Disability	Disable	Whether household head or spouse disabled (yes = 1, no = 0)
Health shock	Health_S	Whether household head or spouse has been hospitalized in the past year (yes = 1, no = 0)
Urban employee medical insurance	Insur_EM	Whether household head or spouse having urban employee medical insurance (yes = 1, no = 0)
Commercial medical insurance	Insur_CM	Whether household head or spouse having commercial medical insurance (yes = 1, no = 0)
Other medical insurance	Insur_OM	Whether household head or spouse having other medical insurance (yes = 1, no = 0)

**Table 2 ijerph-19-04480-t002:** Description results.

Variables	Urban				Rural			
	Min	Max	Mean	SD	Min	Max	Mean	SD
I	2.00	276,451.10	8839.73	12,185.23	2.30	200,985.80	6963.38	10,265.87
ME	1.53	189,987.65	1477.64	4477.59	1.22	200,000.00	1518.05	5037.86
DP	0.00	75,995.06	107.65	1520.75	0.00	74,095.18	66.10	1072.16
LA	0.00	299,230.58	5339.31	17,730.48	0.00	287,356.30	3591.45	12,342.12
Age	29	103	61.33	10.14	19	102	61.94	9.91
Gender	0	1	0.80	0.40	0	1	0.81	0.39
Education	0	19	6.41	4.24	0	19	5.38	4.01
Married	0	1	0.80	0.40	0	1	0.79	0.41
Size	1	18	3.90	1.95	1	15	3.88	1.93
Labor	0	1	0.39	0.32	0	1	0.37	0.32
Area	0	1	0.31	0.46	0	1	0.27	0.45
Health	1	6	3.95	0.92	1	6	4.05	0.95
Diseases	0	1	0.86	0.35	0	1	0.89	0.31
Disable	0	1	0.23	0.42	0	1	0.25	0.43
Health_S	0	1	0.30	0.46	0	1	0.29	0.45
Insur_EM	0	1	0.19	0.39	0	1	0.11	0.32
Insur_CM	0	1	0.04	0.19	0	1	0.03	0.16
Insur_OM	0	1	0.79	0.41	0	6	0.86	0.34

**Table 3 ijerph-19-04480-t003:** Households’ financial vulnerability by health status of household heads.

Group	Urban			Rural		
	2011	2013	2015	2011	2013	2015
Excellent	0.000	0.009	0.057	0.031	0.015	0.046
Very good	0.025	0.014	0.049	0.042	0.015	0.055
Good	0.038	0.021	0.061	0.059	0.031	0.086
Fair	0.051	0.031	0.086	0.079	0.045	0.106
Poor	0.071	0.061	0.133	0.106	0.083	0.150
Very poor	0.099	0.094	0.156	0.127	0.098	0.185

**Table 4 ijerph-19-04480-t004:** Households’ financial vulnerability by labor force proportion of households.

Group	Urban			Rural		
	2011	2013	2015	2011	2013	2015
<0.25	0.078	0.069	0.108	0.119	0.091	0.132
0.25–0.5	0.063	0.046	0.092	0.101	0.059	0.113
0.5–0.75	0.043	0.032	0.054	0.067	0.046	0.062
≥0.75	0.023	0.014	0.054	0.047	0.022	0.070

**Table 5 ijerph-19-04480-t005:** Households’ financial vulnerability by education years of household heads.

Group	Urban			Rural		
	2011	2013	2015	2011	2013	2015
<6	0.112	0.083	0.155	0.136	0.102	0.174
6–9	0.080	0.058	0.119	0.110	0.068	0.133
9–12	0.082	0.042	0.096	0.076	0.048	0.105
12–15	0.080	0.027	0.079	0.066	0.031	0.073
≥15	0.066	0.017	0.047	0.039	0.019	0.053

**Table 6 ijerph-19-04480-t006:** Determinants of financial vulnerability.

Variables	Urban			Rural		
	OLS	FE	RE	OLS	FE	RE
Age	−0.000 **	−0.000	−0.000 **	−0.000 **	−0.001 **	−0.000 **
	(0.000)	(0.002)	(0.000)	(0.000)	(0.000)	(0.000)
Gender	−0.008 **	−0.003	−0.008 **	0.008 **	0.010 *	0.007 **
	(0.002)	(0.005)	(0.002)	(0.002)	(0.005)	(0.002)
Education	−0.005 **	−0.004 **	−0.005 **	−0.006 **	−0.005 **	−0.006 **
	(0.000)	(0.001)	(0.000)	(0.000)	(0.001)	(0.000)
Married	−0.008 **	0.002	−0.008 **	−0.021 **	−0.013 **	−0.020 **
	(0.002)	(0.006)	(0.002)	(0.002)	(0.005)	(0.002)
Size	0.008 **	0.006 **	0.008 **	0.007 **	0.008 **	0.008 **
	(0.001)	(0.001)	(0.001)	(0.001)	(0.001)	(0.001)
Labor	−0.090 **	−0.062 **	−0.091 **	−0.112 **	−0.085 **	−0.112 **
	(0.003)	(0.009)	(0.003)	(0.003)	(0.008)	(0.003)
Area	−0.019 **	−0.020 **	−0.019 **	−0.019 **	−0.004 †	−0.018 **
	(0.001)	(0.005)	(0.001)	(0.001)	(0.004)	(0.001)
Health	0.016 **	0.016 **	0.016 **	0.018 **	0.016 **	0.017 **
	(0.001)	(0.002)	(0.001)	(0.001)	(0.001)	(0.001)
Diseases	0.003	−0.002	0.003	0.007 **	0.008 †	0.008 **
	(0.002)	(0.005)	(0.002)	(0.002)	(0.004)	(0.002)
Disable	0.025 **	0.020 **	0.024 **	0.023 **	0.018 **	0.022 **
	(0.002)	(0.003)	(0.002)	(0.002)	(0.002)	(0.002)
Health_S	0.011 **	0.013 **	0.011 **	0.010 **	0.007 **	0.010 **
	(0.002)	(0.003)	(0.002)	(0.001)	(0.002)	(0.001)
Insur_EM	−0.003 †	−0.023 **	−0.004	−0.011 *	0.001	−0.010 *
	(0.004)	(0.008)	(0.004)	(0.004)	(0.009)	(0.004)
Insur_CM	−0.004	−0.005	−0.004	−0.011 **	−0.012 †	−0.011 **
	(0.003)	(0.006)	(0.003)	(0.004)	(0.007)	(0.004)
Insur_OM	0.005	−0.010	0.004	−0.004	0.001	−0.004
	(0.004)	(0.008)	(0.004)	(0.004)	(0.008)	(0.004)
Constant	0.071	0.030	0.071 **	0.090 **	0.064 **	0.090 **
	(0.008)	(0.018)	(0.008)	(0.008)	(0.016)	(0.008)
R^2^	0.441	0.463	0.435	0.415	0.383	0.366
*p*-value	0.0000	0.0000	0.0000	0.0000	0.0000	0.0000
Number of observations	5559	5559	5559	8718	8718	8718

Note. OLS, FE, RE are ordinary least squares estimator, fixed effect estimator, random effect estimator respectively; standard error in parentheses; † *p* < 0.10, **p* < 0.05, ** *p* < 0.01.

**Table 7 ijerph-19-04480-t007:** Contribution of determinants to financial vulnerability changing for urban households.

Variables	Elasticity	2011–2013			2013–2015		
		Change	Absolute Contribution	Relative Contribution	Change	Absolute Contribution	Relative Contribution
Age	−0.142	1.368	−0.194	1.006	3.351	−0.476	−0.663
Gender	−0.046	23.088	−1.051	5.447	−2.748	0.125	0.174
Education	−0.418	−5.888	2.464	−12.768	−3.557	1.489	2.075
Married	0.021	−3.133	−0.066	0.343	−2.985	−0.063	−0.088
Size	0.419	−8.283	−3.468	17.972	14.593	6.110	8.517
Labor	−0.402	7.552	−3.038	15.742	−9.201	3.701	5.159
Area	−0.103	−3.835	0.394	−2.042	−13.190	1.356	1.890
Health	1.048	0.791	0.829	−4.295	0.202	0.212	0.296
Diseases	−0.031	8.761	−0.273	1.416	−0.458	0.014	0.020
Disable	0.074	−29.391	−2.188	11.339	22.335	1.663	2.318
Health_S	0.096	15.510	1.492	−7.732	21.908	2.108	2.938
Insur_EM	−0.071	156.062	−11.027	57.138	0.047	−0.003	−0.005
Insur_CM	−0.003	57.895	−0.178	0.922	7.692	−0.024	−0.033
Insur_OM	−0.128	−10.130	1.293	−6.700	2.359	−0.301	−0.420
Residual	-	-	−3.968	24.500	-	55.870	79.880
Total	-	-	−19.298	100	-	71.739	100

Note. The elasticity of each variable is equal to the estimated coefficient of fixed effect model multiplied by ($\bar{x}/\bar{y}$). The change of each variable in the table is the actual rate of change multiplied by 100. The absolute contribution of each variable is equal to the elasticity times the change. The sum of the relative contribution of all variables is 100.

**Table 8 ijerph-19-04480-t008:** Contribution of determinants to financial vulnerability changing for rural households.

Variables	Elasticity	2011–2013			2013–2015		
		Change	Absolute Contribution	Relative Contribution	Change	Absolute Contribution	Relative Contribution
Age	−0.453	1.341	−0.608	3.517	3.508	−1.590	−3.043
Gender	0.104	18.575	1.929	−11.161	−2.532	−0.263	−0.503
Education	−0.347	23.102	−8.027	46.444	−2.502	0.869	1.664
Married	−0.125	3.732	−0.467	2.705	−3.238	0.406	0.776
Size	0.395	−9.949	−3.928	22.725	20.818	8.219	15.733
Labor	−0.382	6.085	−2.321	13.431	−18.703	7.136	13.660
Area	−0.013	13.961	−0.181	1.049	−9.153	0.119	0.228
Health	0.781	−2.141	−1.672	9.673	0.746	0.582	1.115
Diseases	0.090	7.351	0.663	−3.836	−1.297	−0.117	−0.224
Disable	0.054	−33.648	−1.810	10.474	16.114	0.867	1.660
Health_S	0.026	4.314	0.114	−0.660	26.316	0.696	1.332
Insur_EM	0.001	206.667	0.198	−1.144	−8.696	−0.008	−0.016
Insur_CM	−0.004	93.750	−0.375	2.169	−6.452	0.026	0.049
Insur_OM	0.014	−5.605	−0.077	0.444	3.207	0.044	0.084
Residual	-	-	−0.721	4.170	-	35.254	67.486
Total	-	-	−17.284	100	-	52.239	100

Note. The elasticity of each variable is equal to the estimated coefficient of fixed effect model multiplied by ($\bar{x}/\bar{y}$). The change of each variable in the table is the actual rate of change multiplied by 100. The absolute contribution of each variable is equal to the elasticity times the change. The sum of the relative contribution of all variables is 100.

## Data Availability

The raw data used in the present study is publicly available at http://charls.pku.edu.cn/index/en.html (accessed on 6 June 2020).

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
