# Peer review of "Household Financial Vulnerability to Income and Medical Expenditure Shocks: Measurement and Determinants"

_ijerph, 2022, doi:10.3390/ijerph19084480_

Round 1

Reviewer 1 Report

The authors must improve the motivation of the research article. Although they are investigating a topic of interest, few national and international studies are cited to motivate the article.
As for section 3. Methodology, no major shortcomings are observed. The authors propose a methodology based on previous research studies. It is presented in a clear and concise way, so this section is correct.
In section 4. Data and Descriptive Statistics, it would be interesting to include in table 1. Determinants of financial vulnerability of households, a new column with the sign of the expected indicator in the calculation of the determinants of financial vulnerability of households ( section 5.2). ). This could be a criterion to accept or reject the validity of the indicator. In fact, the authors argue so in lines 412 to 447.
Finally, sections 6. Discussion and 7. Conclusion repeat the same information. I recommend the authors to delve into the theoretical and practical implications of their results, as well as improve the conclusions, not just summarize the research article.
Finally, I recommend that authors review the citation format of the journal, since in the current format they do not comply with the required format.

Author Response

Dear Reviewer:

Thank you for for the comments concerning our manuscript entitled “Household financial vulnerability to income and medical expenditure shocks: Measurement and determinants” (ijerph-1646362). Those comments are all valuable and very helpful for revising and improving our paper, as well as providing important guiding significance for our research. We have studied the comments carefully and have made corrections which we hope meet with approval. All the revisions to the manuscript have been marked up. The main corrections in the paper and the response to the reviewers’ comments are as follows.
Reviewer 1#

1.Response to comment:The authors must improve the motivation of the research article. Although they are investigating a topic of interest, few national and international studies are cited to motivate the article.

Response: Thanks for this excellent suggestion. We have added the citation of national and international studies in the first paragraph of the introduction (Page 1).
2.Response to comment: Finally, sections 6. Discussion and 7. Conclusion repeat the same information. I recommend the authors to delve into the theoretical and practical implications of their results, as well as improve the conclusions, not just summarize the research article.

Response:It is really true as Reviewer suggested that 7.Conclusion repeats the same information of 6.Discussion. We have revised the section of discussion, which focuses on the highlights of our measure method, the results of household financial vulnerability, and differences in determinants analysis between urban and rural households (Page 15-16).  
3.Response to comment: Finally, I recommend that authors review the citation format of the journal, since in the current format they do not comply with the required format.

Response: We have revised the citation and reference list in the manuscript according to the required format (full text).

Reviewer 2 Report

Referee report on paper entitled “Household financial vulnerability to income and medical expenditure shocks: Measurement and determinants

The paper examines household financial vulnerability by calculating the probability of falling into financial distress under uncertain income and medical expenditure. The authors propose a measure of household financial vulnerability that takes into account not only the income shock but also the medical expenditure shock. They provide an ex-ante indicator of household financial vulnerability considering income and medical expenditure shocks. Abstract contains relevant findings as a result of conducted empirical analysis. The main findings are appealing to the reader.

The research question is very interesting and the paper could potentially contribute to the existing empirical findings. I enjoyed reading the article, it is well written, but authors should address a few issues before the manuscript can be considered for publication in the International Journal of Environmental Research and Public Health. Here are the comments in detail:

  • (Page 8, lines 335-345): The authors note that rural households experience higher financial vulnerability than urban households. Could the authors could provide a formal test of the differences between groups? There are evident differences in income, mean debt repayment and liquid assets, but the differences are not as obvious in health status and demographic variables. Hence, it is necessary to argue the separate analysis in more detail when explaining the descriptive analysis. If authors do not perform some formal tests, then they can explain that the separate analysis is done because there are differences between the groups in terms of their economic-financial situation.
  • (Page 9, lines 353-357): The authors provide estimates for the income and medical expenditure equations. The independent variables in the medical expenditure model are not exactly the same as in the income regression model. Why does the income model not use the same set of independent variables as the medical expenditure model?
  • (Page 11-12 – Table 6 and lines 412-432): R2 is an appropriate measure of fit only for OLS (in the panel data set it is Pooled OLS – POLS). For FE and RE only Hausman test should be reported, in which authors provide an evidence that FE is appropriate model. Perhaps it makes more sense to report the number of household in the sample (number of id’s) instead of the P-value (last line of the Table 6). When it comes to explaining the results of the FE model, the authors should explain the results in terms of significance level. There is a lack of explanation for rural variables (line 417 should explained in more detail - all demographic (and later health) variables are significant at the 5% level, except for area, which is significant at 10% significance level. The authors should also highlight the differences between obtained the results for urban and rural households. In urban households, age, gender, married and diseases are not significant as in rural households. There is also no explanation for gender – what does it mean (positive and significant for rural households). Then, marital status of rural households has a significant negative impact on financial vulnerability…why?

IMPORTANT! There are some issues with the results from FE model for the variable Commerciale Insurance (Insur_CM for Urban households). Table 6 shows that the Insur_CM variable (POLS and FE - Urban households) is significant at the 5% significance level (POLS) and the 10% significance level (FE). This cannot be true because dividing coefficient with standard error does not exceed 1.96 and 1.67, respectively.

  • (Page 12, Table 7): The variable Age should not be bold and underlined.
  • (Discussion and conclusion) are too similar. I recommend combining Discussion and Conclusion in one chapter to avoid repetition.
  • Since the authors found that rural households in China are more financially vulnerable than urban households, the authors should provide recommendations for policy makers when it comes to rural areas.

Author Response

Dear Reviewer:

Thank you for the comments concerning our manuscript entitled “Household financial vulnerability to income and medical expenditure shocks: Measurement and determinants” (ijerph-1646362). Those comments are all valuable and very helpful for revising and improving our paper, as well as providing important guiding significance for our research. We have studied the comments carefully and have made corrections which we hope meet with approval. All the revisions to the manuscript have been marked up. The main corrections in the paper and the response to the reviewers’ comments are as follows.
Reviewer 2#

1.Response to comment: (Page 8, lines 335-345): The authors note that rural households experience higher financial vulnerability than urban households. Could the authors could provide a formal test of the differences between groups? There are evident differences in income, mean debt repayment and liquid assets, but the differences are not as obvious in health status and demographic variables. Hence, it is necessary to argue the separate analysis in more detail when explaining the descriptive analysis. If authors do not perform some formal tests, then they can explain that the separate analysis is done because there are differences between the groups in terms of their economic-financial situation.

Response: There are considerable differences between urban and rural areas in China because Chinese urban areas experience faster economic development than rural ones. The descriptive statistics analysis is performed to show the differences on Pages 8-9. The average values of these economic variables show that there is a significant difference in economic-financial situations between urban and rural areas.

2.Response to comment: (Page 9, lines 353-357): The authors provide estimates for the income and medical expenditure equations. The independent variables in the medical expenditure model are not exactly the same as in the income regression model. Why does the income model not use the same set of independent variables as the medical expenditure model?

Response: As a risk management tool, medical insurance could reduce the loss of health shocks, but not directly affect the income of household. Therefore, The medical insurance variables including insur_EM, insur_CM, and insur_OM, are also taken into account in the medical expenditure model, while are not included in the income regression model.

3.Response to comment:(Page 11-12 – Table 6 and lines 412-432): R2 is an appropriate measure of fit only for OLS (in the panel data set it is Pooled OLS – POLS). For FE and RE only Hausman test should be reported, in which authors provide an evidence that FE is appropriate model. Perhaps it makes more sense to report the number of household in the sample (number of id’s) instead of the P-value (last line of the Table 6).

Response: Thanks for this excellent suggestion. We have added the number of observations in the last line of Table 6.

4.Response to comment:When it comes to explaining the results of the FE model, the authors should explain the results in terms of significance level. There is a lack of explanation for rural variables (line 417 should explained in more detail - all demographic (and later health) variables are significant at the 5% level, except for area, which is significant at 10% significance level. The authors should also highlight the differences between obtained the results for urban and rural households. In urban households, age, gender, married and diseases are not significant as in rural households. There is also no explanation for gender – what does it mean (positive and significant for rural households). Then, marital status of rural households has a significant negative impact on financial vulnerability…why?

Response: Thanks for this suggestion. We have added explanations about the difference between obtained the results for urban and rural households in the section of discussion (Page 21-22).

5.Response to comment:IMPORTANT! There are some issues with the results from FE model for the variable Commerciale Insurance (Insur_CM for Urban households). Table 6 shows that the Insur_CM variable (POLS and FE - Urban households) is significant at the 5% significance level (POLS) and the 10% significance level (FE). This cannot be true because dividing coefficient with standard error does not exceed 1.96 and 1.67, respectively.

Response: We are very sorry for this error flag about the significance of Insur_CM variable. We have revised it in Table 6 and the corresponding expression of the results in the text.  

6.Response to comment:(Page 12, Table 7): The variable Age should not be bold and underlined.

Response:We have revised it in Table 7 (Page 19).

7.Response to comment:(Discussion and conclusion) are too similar. I recommend combining Discussion and Conclusion in one chapter to avoid repetition.

Response:It is really true as Reviewer suggested that there are some repetition in Discussion and Conclusion. We have revised the section of discussion, which focuses on the highlights of our measure method, the results of household financial vulnerability, and differences in determinants analysis between urban and rural households (Page 15-16). 

8.Response to comment:Since the authors found that rural households in China are more financially vulnerable than urban households, the authors should provide recommendations for policy makers when it comes to rural areas.

Response:Thanks for this excellent suggestion. We have added some policy recommendations for rural areas about medical insurance for rural households in the section of Conclusion (Page 17).

We appreciate for your warm work earnestly, and hope that the correction will meet with approval. Once again, thank you very much for your comments and suggestions.

Thank you and best regards.
Yours sincerely,
Lei He

Reviewer 3 Report

In this article, Authors presented their own method of measuring household vulnerability in relation to income shocks and unexpected health expenditure. This method was used to measure household sensitivity in China in relation to urban and rural households. The structure of the text is clear and the subsequent research points are exhaustively presented. Own definition of sensitivity was proposed, measures for its measurement were constructed, and determinants of sensitivity were selected and tested.

The authors made a proper literature review for the three above research points. Due to the lack of a commonly accepted definition, they proposed their own and following this line of reasoning, they constructed measures that they used for calculations based on data from China. In addition, they examined the determinants of the obtained results and clearly interpreted the entire study and results.

In the face of the global economic and humanitarian crisis, the research problem undertaken by the authors is particularly important not only for financial institutions that grant loans. Thanks to the results, health politicians gained knowledge that can be used to improve the functioning of the health system.

Research creativity should be emphasized, thanks to which the proposed research method contributes to the literature.

Author Response

Dear Reviewer:

Thank you for the comments concerning our manuscript entitled “Household financial vulnerability to income and medical expenditure shocks: Measurement and determinants” (ijerph-1646362). Those comments are all valuable and very helpful for revising and improving our paper, as well as providing important guiding significance for our research. We have studied the comments carefully and have made corrections which we hope meet with approval. All the revisions to the manuscript have been marked up. The main corrections in the paper and the response to the reviewers’ comments are as follows.
Reviewer 3#

1.Response to comment: 1 / It is worth comparing the obtained results of the financial vulnerability of households in China with the results of other researchers concerning China: - / ex-ante (analogous to that adopted in the study) - / ex-post and comparing the results, which would allow to verify the effectiveness of the measurement method proposed by the authors. The comparison should take into account the diversity of assumptions and tools used for calculations in individual studies.

Response: Although the issue of household financial vulnerability in some countries (e.g. Indonesia, Chile, Pakistan, Italy) is frequently investigated, it is not studied in China. Thus, we can not compare the obtained results of our work with the results of other researchers concerning China. However, we argue theoretically the the advantages of our measure in the section of discussion. The highlights of our measure method are manifested in two aspects. On the one hand, it can reflect the capacity of household to deal with income and medical expenditure shocks. On the other hand, the probability of falling into financial distress can quantify the degree of household financial vulnerability beforehand, which helps to assess the financial risks of the household sector.

2.Response to comment:2 / Taking into account the pitfalls that always lie in the comparisons of studies conducted in different countries, it is worth comparing the results and conclusions obtained in the conducted study with the reasonably selected foreign studies. It would be interesting to compare the effectiveness of the ex-ante and ex-post measures of the conducted study with the effectiveness of selected studies from other countries.

Response:Thanks for this excellent suggestion. In the new version, we have added some comparisons to the existing foreign studies in the analysis of determinants (Page 12). The further study of our work will focus on comparing the effectiveness of the ex-ante and ex-post measures.

  1. 3.Response to comment:3/ The literature review includes objective and subjective measures, while the division into ex-ante and ex-post measures has been omitted, and it is worth presenting them.

Response: Most objective and subjective measures are ex-post measures. The ex-ante measure of household vulnerability is mainly applied to investigate the vulnerability of households to poverty by the probability method [13-14]. Few ex-ante measures of household vulnerability are applied in empirical analysis, e.g. [15]. We have added it in the section of literature review (Page 5).

4.Response to comment: Thanks to the literature presenting ex-ante measures, it is worth commenting on the differences between the tools and functions used for calculations. These hidden elements of the study determine the obtained results, which are compared.

Response:We have improved our argument about the differences between our measure and ex-ante measures proposed by existing literature (Page 3). Differing from the ex-ante measure of [15], the heteroscedasticity of income and medical expenditure shocks for different characteristics of households is included in our measure, and it is estimated by a three-step feasible generalized least squares (FGLS) procedure. The comparison between our measure and ex-ante measure of [13-14] is shown in the previous version of the manuscript.

We appreciate for your warm work earnestly, and hope that the correction will meet with approval. Once again, thank you very much for your comments and suggestions.

Thank you and best regards.
Yours sincerely,
Lei He

Round 2

Reviewer 1 Report

Dear authors of the paper entitled "Household financial vulnerability to income and medical expenditure shocks: Measurement and determinants". After reading the incorporated modifications, I consider that your paper meets all the formal requirements, as well as with sufficient scientific quality, to be recommended for publication.
I congratulate you, because I consider that you have proposed a novel, relevant and original paper for the generation of new scientific knowledge in the field of financial inclusion.